# XLNet: Generalized Autoregressive Pretraining for Language Understanding

**Zhilin Yang**[*1], **Zihang Dai**[*12], **Yiming Yang**[1], **Jaime Carbonell**[1],
**Ruslan Salakhutdinov**[1], **Quoc V. Le**[2]
[1]Carnegie Mellon University, [2]Google AI Brain Team
{zhiliny,dzihang,yiming,jgc,rsalakhu}@cs.cmu.edu, qvl@google.com

## Abstract

With the capability of modeling bidirectional contexts, denoising autoencoding based pretraining like BERT achieves better performance than pretraining approaches based on autoregressive language modeling. However, relying on corrupting the input with masks, BERT neglects dependency between the masked positions and suffers from a pretrain-finetune discrepancy. In light of these pros and cons, we propose XLNet, a generalized autoregressive pretraining method that (1) enables learning bidirectional contexts by maximizing the expected likelihood over all permutations of the factorization order and (2) overcomes the limitations of BERT thanks to its autoregressive formulation. Furthermore, XLNet integrates ideas from Transformer-XL, the state-of-the-art autoregressive model, into pretraining. Empirically, under comparable experiment setting, XLNet outperforms BERT on 20 tasks, often by a large margin, including question answering, natural language inference, sentiment analysis, and document ranking.[1].

## 1 Introduction

Unsupervised representation learning has been highly successful in the domain of natural language processing [7, 22, 27, 28, 10]. Typically, these methods first pretrain neural networks on large-scale unlabeled text corpora, and then finetune the models or representations on downstream tasks. Under this shared high-level idea, different unsupervised pretraining objectives have been explored in literature. Among them, autoregressive (AR) language modeling and autoencoding (AE) have been the two most successful pretraining objectives.

AR language modeling seeks to estimate the probability distribution of a text corpus with an autoregressive model [7, 27, 28]. Specifically, given a text sequence $\mathbf{x} = (x_1, \cdots, x_T)$, AR language modeling factorizes the likelihood into a forward product $p(\mathbf{x}) = \prod_{t=1}^{T} p(x_t \mid \mathbf{x}_{<t})$ or a backward one $p(\mathbf{x}) = \prod_{t=T}^{1} p(x_t \mid \mathbf{x}_{>t})$. A parametric model (e.g. a neural network) is trained to model each conditional distribution. Since an AR language model is only trained to encode a uni-directional context (either forward or backward), it is not effective at modeling deep bidirectional contexts. On the contrary, downstream language understanding tasks often require bidirectional context information. This results in a gap between AR language modeling and effective pretraining.

In comparison, AE based pretraining does not perform explicit density estimation but instead aims to reconstruct the original data from corrupted input. A notable example is BERT [10], which has been the state-of-the-art pretraining approach. Given the input token sequence, a certain portion of tokens are replaced by a special symbol [MASK], and the model is trained to recover the original tokens from the corrupted version. Since density estimation is not part of the objective, BERT is allowed to utilize

---

[*]Equal contribution. Order determined by swapping the one in [9].
[1]Pretrained models and code are available at `https://github.com/zihangdai/xlnet`

bidirectional contexts for reconstruction. As an immediate benefit, this closes the aforementioned bidirectional information gap in AR language modeling, leading to improved performance. However, the artificial symbols like [MASK] used by BERT during pretraining are absent from real data at finetuning time, resulting in a pretrain-finetune discrepancy. Moreover, since the predicted tokens are masked in the input, BERT is not able to model the joint probability using the product rule as in AR language modeling. In other words, BERT assumes the predicted tokens are independent of each other given the unmasked tokens, which is oversimplified as high-order, long-range dependency is prevalent in natural language [9].

Faced with the pros and cons of existing language pretraining objectives, in this work, we propose XLNet, a generalized autoregressive method that leverages the best of both AR language modeling and AE while avoiding their limitations.

- Firstly, instead of using a fixed forward or backward factorization order as in conventional AR models, XLNet maximizes the expected log likelihood of a sequence w.r.t. **all possible permutations of the factorization order**. Thanks to the permutation operation, the context for each position can consist of tokens from both left and right. In expectation, each position learns to utilize contextual information from all positions, i.e., capturing bidirectional context.
- Secondly, as a generalized AR language model, XLNet does not rely on data corruption. Hence, XLNet does not suffer from the pretrain-finetune discrepancy that BERT is subject to. Meanwhile, the autoregressive objective also provides a natural way to use the product rule for factorizing the joint probability of the predicted tokens, eliminating the independence assumption made in BERT.

In addition to a novel pretraining objective, XLNet improves architectural designs for pretraining.

- Inspired by the latest advancements in AR language modeling, XLNet integrates the segment recurrence mechanism and relative encoding scheme of Transformer-XL [9] into pretraining, which empirically improves the performance especially for tasks involving a longer text sequence.
- Naively applying a Transformer(-XL) architecture to permutation-based language modeling does not work because the factorization order is arbitrary and the target is ambiguous. As a solution, we propose to reparameterize the Transformer(-XL) network to remove the ambiguity.

Empirically, under comparable experiment setting, XLNet consistently outperforms BERT [10] on a wide spectrum of problems including GLUE language understanding tasks, reading comprehension tasks like SQuAD and RACE, text classification tasks such as Yelp and IMDB, and the ClueWeb09-B document ranking task.

**Related Work** The idea of permutation-based AR modeling has been explored in [32, 12], but there are several key differences. Firstly, previous models aim to improve density estimation by baking an "orderless" inductive bias into the model while XLNet is motivated by enabling AR language models to learn bidirectional contexts. Technically, to construct a valid target-aware prediction distribution, XLNet incorporates the target position into the hidden state via two-stream attention while previous permutation-based AR models relied on implicit position awareness inherent to their MLP architectures. Finally, for both orderless NADE and XLNet, we would like to emphasize that "orderless" does not mean that the input sequence can be randomly permuted but that the model allows for different factorization orders of the distribution.

Another related idea is to perform autoregressive denoising in the context of text generation [11], which only considers a fixed order though.

## 2 Proposed Method

### 2.1 Background

In this section, we first review and compare the conventional AR language modeling and BERT for language pretraining. Given a text sequence $\mathbf{x} = [x_1, \cdots, x_T]$, AR language modeling performs pretraining by maximizing the likelihood under the forward autoregressive factorization:

$$\max_{\theta} \quad \log p_\theta(\mathbf{x}) = \sum_{t=1}^{T} \log p_\theta(x_t \mid \mathbf{x}_{<t}) = \sum_{t=1}^{T} \log \frac{\exp\left(h_\theta(\mathbf{x}_{1:t-1})^\top e(x_t)\right)}{\sum_{x'} \exp\left(h_\theta(\mathbf{x}_{1:t-1})^\top e(x')\right)}, \qquad (1)$$

where $h_\theta(\mathbf{x}_{1:t-1})$ is a context representation produced by neural models, such as RNNs or Transformers, and $e(x)$ denotes the embedding of $x$. In comparison, BERT is based on denoising auto-encoding. Specifically, for a text sequence $\mathbf{x}$, BERT first constructs a corrupted version $\hat{\mathbf{x}}$ by randomly setting a portion (e.g. 15%) of tokens in $\mathbf{x}$ to a special symbol [MASK]. Let the masked tokens be $\bar{\mathbf{x}}$. The training objective is to reconstruct $\bar{\mathbf{x}}$ from $\hat{\mathbf{x}}$:

$$\max_\theta \quad \log p_\theta(\bar{\mathbf{x}} \mid \hat{\mathbf{x}}) \approx \sum_{t=1}^T m_t \log p_\theta(x_t \mid \hat{\mathbf{x}}) = \sum_{t=1}^T m_t \log \frac{\exp\left(H_\theta(\hat{\mathbf{x}})_t^\top e(x_t)\right)}{\sum_{x'} \exp\left(H_\theta(\hat{\mathbf{x}})_t^\top e(x')\right)}, \quad (2)$$

where $m_t = 1$ indicates $x_t$ is masked, and $H_\theta$ is a Transformer that maps a length-$T$ text sequence $\mathbf{x}$ into a sequence of hidden vectors $H_\theta(\mathbf{x}) = [H_\theta(\mathbf{x})_1, H_\theta(\mathbf{x})_2, \cdots, H_\theta(\mathbf{x})_T]$. The pros and cons of the two pretraining objectives are compared in the following aspects:

- **Independence Assumption**: As emphasized by the $\approx$ sign in Eq. (2), BERT factorizes the joint conditional probability $p(\bar{\mathbf{x}} \mid \hat{\mathbf{x}})$ based on an independence assumption that all masked tokens $\bar{\mathbf{x}}$ are separately reconstructed. In comparison, the AR language modeling objective (1) factorizes $p_\theta(\mathbf{x})$ using the product rule that holds universally without such an independence assumption.
- **Input noise**: The input to BERT contains artificial symbols like [MASK] that never occur in downstream tasks, which creates a pretrain-finetune discrepancy. Replacing [MASK] with original tokens as in [10] does not solve the problem because original tokens can be only used with a small probability — otherwise Eq. (2) will be trivial to optimize. In comparison, AR language modeling does not rely on any input corruption and does not suffer from this issue.
- **Context dependency**: The AR representation $h_\theta(\mathbf{x}_{1:t-1})$ is only conditioned on the tokens up to position $t$ (i.e. tokens to the left), while the BERT representation $H_\theta(\mathbf{x})_t$ has access to the contextual information on both sides. As a result, the BERT objective allows the model to be pretrained to better capture bidirectional context.

## 2.2 Objective: Permutation Language Modeling

According to the comparison above, AR language modeling and BERT possess their unique advantages over the other. A natural question to ask is whether there exists a pretraining objective that brings the advantages of both while avoiding their weaknesses.

Borrowing ideas from orderless NADE [32], we propose the permutation language modeling objective that not only retains the benefits of AR models but also allows models to capture bidirectional contexts. Specifically, for a sequence $\mathbf{x}$ of length $T$, there are $T!$ different orders to perform a valid autoregressive factorization. Intuitively, if model parameters are shared across all factorization orders, in expectation, the model will learn to gather information from all positions on both sides.

To formalize the idea, let $\mathcal{Z}_T$ be the set of all possible permutations of the length-$T$ index sequence $[1, 2, \ldots, T]$. We use $z_t$ and $\mathbf{z}_{<t}$ to denote the $t$-th element and the first $t-1$ elements of a permutation $\mathbf{z} \in \mathcal{Z}_T$. Then, our proposed permutation language modeling objective can be expressed as follows:

$$\max_\theta \quad \mathbb{E}_{\mathbf{z} \sim \mathcal{Z}_T}\left[\sum_{t=1}^T \log p_\theta(x_{z_t} \mid \mathbf{x}_{\mathbf{z}_{<t}})\right]. \quad (3)$$

Essentially, for a text sequence $\mathbf{x}$, we sample a factorization order $\mathbf{z}$ at a time and decompose the likelihood $p_\theta(\mathbf{x})$ according to factorization order. Since the same model parameter $\theta$ is shared across all factorization orders during training, in expectation, $x_t$ has seen every possible element $x_i \neq x_t$ in the sequence, hence being able to capture the bidirectional context. Moreover, as this objective fits into the AR framework, it naturally avoids the independence assumption and the pretrain-finetune discrepancy discussed in Section 2.1.

**Remark on Permutation** The proposed objective only permutes the factorization order, not the sequence order. In other words, we keep the original sequence order, use the positional encodings corresponding to the original sequence, and rely on a proper attention mask in Transformers to achieve permutation of the factorization order. Note that this choice is necessary, since the model will only encounter text sequences with the natural order during finetuning.

To provide an overall picture, we show an example of predicting the token $x_3$ given the same input sequence $\mathbf{x}$ but under different factorization orders in the Appendix A.7 with Figure 4.

## 2.3 Architecture: Two-Stream Self-Attention for Target-Aware Representations

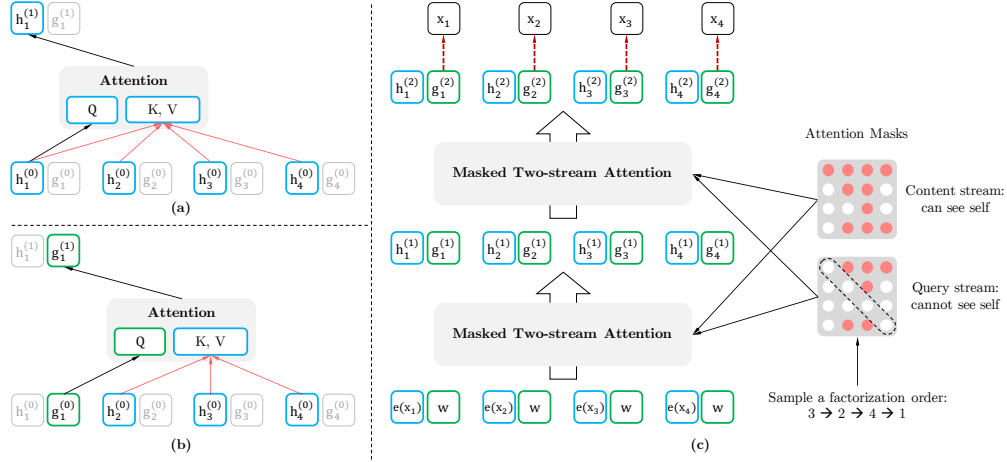

Figure 1: (a): Content stream attention, which is the same as the standard self-attention. (b): Query stream attention, which does not have access information about the content $x_{z_t}$. (c): Overview of the permutation language modeling training with two-stream attention.

While the permutation language modeling objective has desired properties, naive implementation with standard Transformer parameterization may not work. To see the problem, assume we parameterize the next-token distribution $p_\theta(X_{z_t} \mid \mathbf{x}_{\mathbf{z}_{<t}})$ using the standard Softmax formulation, i.e., $p_\theta(X_{z_t} = x \mid \mathbf{x}_{\mathbf{z}_{<t}}) = \frac{\exp\left(e(x)^\top h_\theta(\mathbf{x}_{\mathbf{z}_{<t}})\right)}{\sum_{x'} \exp\left(e(x')^\top h_\theta(\mathbf{x}_{\mathbf{z}_{<t}})\right)}$, where $h_\theta(\mathbf{x}_{\mathbf{z}_{<t}})$ denotes the hidden representation of $\mathbf{x}_{\mathbf{z}_{<t}}$ produced by the shared Transformer network after proper masking. Now notice that the representation $h_\theta(\mathbf{x}_{\mathbf{z}_{<t}})$ does not depend on which position it will predict, i.e., the value of $z_t$. Consequently, the same distribution is predicted regardless of the target position, which is not able to learn useful representations (see Appendix A.1 for a concrete example). To avoid this problem, we propose to re-parameterize the next-token distribution to be target position aware:

$$p_\theta(X_{z_t} = x \mid \mathbf{x}_{\mathbf{z}_{<t}}) = \frac{\exp\left(e(x)^\top g_\theta(\mathbf{x}_{\mathbf{z}_{<t}}, z_t)\right)}{\sum_{x'} \exp\left(e(x')^\top g_\theta(\mathbf{x}_{\mathbf{z}_{<t}}, z_t)\right)}, \tag{4}$$

where $g_\theta(\mathbf{x}_{\mathbf{z}_{<t}}, z_t)$ denotes a new type of representations which additionally take the target position $z_t$ as input.

**Two-Stream Self-Attention** While the idea of target-aware representations removes the ambiguity in target prediction, how to formulate $g_\theta(\mathbf{x}_{\mathbf{z}_{<t}}, z_t)$ remains a non-trivial problem. Among other possibilities, we propose to "stand" at the target position $z_t$ and rely on the position $z_t$ to gather information from the context $\mathbf{x}_{\mathbf{z}_{<t}}$ through attention. For this parameterization to work, there are two requirements that are contradictory in a standard Transformer architecture: (1) to predict the token $x_{z_t}$, $g_\theta(\mathbf{x}_{\mathbf{z}_{<t}}, z_t)$ should only use the *position* $z_t$ and not the *content* $x_{z_t}$, otherwise the objective becomes trivial; (2) to predict the other tokens $x_{z_j}$ with $j > t$, $g_\theta(\mathbf{x}_{\mathbf{z}_{<t}}, z_t)$ should also encode the content $x_{z_t}$ to provide full contextual information. To resolve such a contradiction, we propose to use two sets of hidden representations instead of one:

- The content representation $h_\theta(\mathbf{x}_{\mathbf{z}_{\leq t}})$, or abbreviated as $h_{z_t}$, which serves a similar role to the standard hidden states in Transformer. This representation encodes *both* the context and $x_{z_t}$ itself.
- The query representation $g_\theta(\mathbf{x}_{\mathbf{z}_{<t}}, z_t)$, or abbreviated as $g_{z_t}$, which only has access to the contextual information $\mathbf{x}_{\mathbf{z}_{<t}}$ and the position $z_t$, but not the content $x_{z_t}$, as discussed above.

Computationally, the first layer query stream is initialized with a trainable vector, i.e. $g_i^{(0)} = w$, while the content stream is set to the corresponding word embedding, i.e. $h_i^{(0)} = e(x_i)$. For each self-attention layer $m = 1, \ldots, M$, the two streams of representations are *schematically*[2] updated

with a shared set of parameters as follows (illustrated in Figures 1 (a) and (b)):

$$g_{z_t}^{(m)} \leftarrow \text{Attention}(\mathbf{Q} = g_{z_t}^{(m-1)}, \text{KV} = \mathbf{h}_{\mathbf{z}_{<t}}^{(m-1)}; \theta), \quad \text{(query stream: use } z_t \text{ but cannot see } x_{z_t})$$

$$h_{z_t}^{(m)} \leftarrow \text{Attention}(\mathbf{Q} = h_{z_t}^{(m-1)}, \text{KV} = \mathbf{h}_{\mathbf{z}_{\leq t}}^{(m-1)}; \theta), \quad \text{(content stream: use both } z_t \text{ and } x_{z_t}).$$

where Q, K, V denote the query, key, and value in an attention operation [33]. The update rule of the content representations is exactly the same as the standard self-attention, so during finetuning, we can simply drop the query stream and use the content stream as a normal Transformer(-XL). Finally, we can use the last-layer query representation $g_{z_t}^{(M)}$ to compute Eq. (4).

**Partial Prediction** While the permutation language modeling objective (3) has several benefits, it is a much more challenging optimization problem due to the permutation and causes slow convergence in preliminary experiments. To reduce the optimization difficulty, we choose to only predict the last tokens in a factorization order. Formally, we split $\mathbf{z}$ into a non-target subsequence $\mathbf{z}_{\leq c}$ and a target subsequence $\mathbf{z}_{>c}$, where $c$ is the cutting point. The objective is to maximize the log-likelihood of the target subsequence conditioned on the non-target subsequence, i.e.,

$$\max_{\theta} \quad \mathbb{E}_{\mathbf{z} \sim \mathcal{Z}_T} \left[ \log p_\theta(\mathbf{x}_{\mathbf{z}_{>c}} \mid \mathbf{x}_{\mathbf{z}_{\leq c}}) \right] = \mathbb{E}_{\mathbf{z} \sim \mathcal{Z}_T} \left[ \sum_{t=c+1}^{|\mathbf{z}|} \log p_\theta(x_{z_t} \mid \mathbf{x}_{\mathbf{z}_{<t}}) \right]. \tag{5}$$

Note that $\mathbf{z}_{>c}$ is chosen as the target because it possesses the longest context in the sequence given the current factorization order $\mathbf{z}$. A hyperparameter $K$ is used such that about $1/K$ tokens are selected for predictions; i.e., $|\mathbf{z}| / (|\mathbf{z}| - c) \approx K$. For unselected tokens, their query representations need not be computed, which saves speed and memory.

## 2.4 Incorporating Ideas from Transformer-XL

Since our objective function fits in the AR framework, we incorporate the state-of-the-art AR language model, Transformer-XL [9], into our pretraining framework, and name our method after it. We integrate two important techniques in Transformer-XL, namely the relative positional encoding scheme and the segment recurrence mechanism. We apply relative positional encodings based on the original sequence as discussed earlier, which is straightforward. Now we discuss how to integrate the recurrence mechanism into the proposed permutation setting and enable the model to reuse hidden states from previous segments. Without loss of generality, suppose we have two segments taken from a long sequence $\mathbf{s}$; i.e., $\tilde{\mathbf{x}} = \mathbf{s}_{1:T}$ and $\mathbf{x} = \mathbf{s}_{T+1:2T}$. Let $\tilde{\mathbf{z}}$ and $\mathbf{z}$ be permutations of $[1 \cdots T]$ and $[T+1 \cdots 2T]$ respectively. Then, based on the permutation $\tilde{\mathbf{z}}$, we process the first segment, and then cache the obtained content representations $\tilde{\mathbf{h}}^{(m)}$ for each layer $m$. Then, for the next segment $\mathbf{x}$, the attention update with memory can be written as

$$h_{z_t}^{(m)} \leftarrow \text{Attention}(\mathbf{Q} = h_{z_t}^{(m-1)}, \text{KV} = \left[ \tilde{\mathbf{h}}^{(m-1)}, \mathbf{h}_{\mathbf{z}_{\leq t}}^{(m-1)} \right]; \theta)$$

where $[., .]$ denotes concatenation along the sequence dimension. Notice that positional encodings only depend on the actual positions in the original sequence. Thus, the above attention update is independent of $\tilde{\mathbf{z}}$ once the representations $\tilde{\mathbf{h}}^{(m)}$ are obtained. This allows caching and reusing the memory without knowing the factorization order of the previous segment. In expectation, the model learns to utilize the memory over all factorization orders of the last segment. The query stream can be computed in the same way. Finally, Figure 1 (c) presents an overview of the proposed permutation language modeling with two-stream attention (see Appendix A.7 for more detailed illustration).

## 2.5 Modeling Multiple Segments

Many downstream tasks have multiple input segments, e.g., a question and a context paragraph in question answering. We now discuss how we pretrain XLNet to model multiple segments in the autoregressive framework. During the pretraining phase, following BERT, we randomly sample two segments (either from the same context or not) and treat the concatenation of two segments as one sequence to perform permutation language modeling. We only reuse the memory that belongs to the same context. Specifically, the input to our model is the same as BERT: [CLS, A, SEP, B, SEP], where "SEP" and "CLS" are two special symbols and "A" and "B" are the two segments. Although

we follow the two-segment data format, XLNet-Large does not use the objective of next sentence prediction [10] as it does not show consistent improvement in our ablation study (see Section 3.4).

**Relative Segment Encodings** Architecturally, different from BERT that adds an absolute segment embedding to the word embedding at each position, we extend the idea of relative encodings from Transformer-XL to also encode the segments. Given a pair of positions $i$ and $j$ in the sequence, if $i$ and $j$ are from the same segment, we use a segment encoding $\mathbf{s}_{ij} = \mathbf{s}_+$ or otherwise $\mathbf{s}_{ij} = \mathbf{s}_-$, where $\mathbf{s}_+$ and $\mathbf{s}_-$ are learnable model parameters for each attention head. In other words, we only consider whether the two positions are *within the same segment*, as opposed to considering *which specific segments they are from*. This is consistent with the core idea of relative encodings; i.e., only modeling the relationships between positions. When $i$ attends to $j$, the segment encoding $\mathbf{s}_{ij}$ is used to compute an attention weight $a_{ij} = (\mathbf{q}_i + \mathbf{b})^\top \mathbf{s}_{ij}$, where $\mathbf{q}_i$ is the query vector as in a standard attention operation and $\mathbf{b}$ is a learnable head-specific bias vector. Finally, the value $a_{ij}$ is added to the normal attention weight. There are two benefits of using relative segment encodings. First, the inductive bias of relative encodings improves generalization [9]. Second, it opens the possibility of finetuning on tasks that have more than two input segments, which is not possible using absolute segment encodings.

## 2.6 Discussion

Comparing Eq. (2) and (5), we observe that both BERT and XLNet perform partial prediction, i.e., only predicting a subset of tokens in the sequence. This is a necessary choice for BERT because if all tokens are masked, it is impossible to make any meaningful predictions. In addition, for both BERT and XLNet, partial prediction plays a role of reducing optimization difficulty by only predicting tokens with sufficient context. However, the independence assumption discussed in Section 2.1 disables BERT to model dependency between targets.

To better understand the difference, let's consider a concrete example [New, York, is, a, city]. Suppose both BERT and XLNet select the two tokens [New, York] as the prediction targets and maximize $\log p(\text{New York} \mid \text{is a city})$. Also suppose that XLNet samples the factorization order [is, a, city, New, York]. In this case, BERT and XLNet respectively reduce to the following objectives:

$$\mathcal{J}_{\text{BERT}} = \log p(\text{New} \mid \text{is a city}) + \log p(\text{York} \mid \text{is a city}),$$

$$\mathcal{J}_{\text{XLNet}} = \log p(\text{New} \mid \text{is a city}) + \log p(\text{York} \mid \text{New}, \text{is a city}).$$

Notice that XLNet is able to capture the dependency between the pair (New, York), which is omitted by BERT. Although in this example, BERT learns some dependency pairs such as (New, city) and (York, city), it is obvious that XLNet always learns **more** dependency pairs given the same target and contains "denser" effective training signals.

For more formal analysis and further discussion, please refer to Appendix A.5.

# 3 Experiments

## 3.1 Pretraining and Implementation

Following BERT [10], we use the BooksCorpus [40] and English Wikipedia as part of our pretraining data, which have 13GB plain text combined. In addition, we include Giga5 (16GB text) [26], ClueWeb 2012-B (extended from [5]), and Common Crawl [6] for pretraining. We use heuristics to aggressively filter out short or low-quality articles for ClueWeb 2012-B and Common Crawl, which results in 19GB and 110GB text respectively. After tokenization with SentencePiece [17], we obtain 2.78B, 1.09B, 4.75B, 4.30B, and 19.97B subword pieces for Wikipedia, BooksCorpus, Giga5, ClueWeb, and Common Crawl respectively, which are 32.89B in total.

Our largest model XLNet-Large has the same architecture hyperparameters as BERT-Large, which results in a similar model size. During pretraining, we always use a full sequence length of 512. Firstly, to provide a fair comparison with BERT (section 3.2), we also trained XLNet-Large-wikibooks on BooksCorpus and Wikipedia only, where we reuse all pretraining hyper-parameters as in the original BERT. Then, we scale up the training of XLNet-Large by using all the datasets described above. Specifically, we train on 512 TPU v3 chips for 500K steps with an Adam weight decay optimizer, linear learning rate decay, and a batch size of 8192, which takes about 5.5 days. It was

observed that the model still underfits the data at the end of training. Finally, we perform ablation study (section 3.4) based on the XLNet-Base-wikibooks.

Since the recurrence mechanism is introduced, we use a bidirectional data input pipeline where each of the forward and backward directions takes half of the batch size. For training XLNet-Large, we set the partial prediction constant $K$ as 6 (see Section 2.3). Our finetuning procedure follows BERT [10] except otherwise specified[3]. We employ an idea of *span-based prediction*, where we first sample a length $L \in [1, \cdots, 5]$, and then randomly select a consecutive span of $L$ tokens as prediction targets within a context of $(KL)$ tokens.

We use a variety of natural language understanding datasets to evaluate the performance of our method. Detailed descriptions of the settings for all the datasets can be found in Appendix A.3.

## 3.2 Fair Comparison with BERT

| Model | SQuAD1.1 | SQuAD2.0 | RACE | MNLI | QNLI | QQP | RTE | SST-2 | MRPC | CoLA | STS-B |
|---|---|---|---|---|---|---|---|---|---|---|---|
| BERT-Large (Best of 3) | 86.7/92.8 | 82.8/85.5 | 75.1 | 87.3 | 93.0 | 91.4 | 74.0 | 94.0 | 88.7 | 63.7 | 90.2 |
| XLNet-Large-wikibooks | 88.2/94.0 | 85.1/87.8 | 77.4 | 88.4 | 93.9 | 91.8 | 81.2 | 94.4 | 90.0 | 65.2 | 91.1 |

Table 1: Fair comparison with BERT. All models are trained using the same data and hyperparameters as in BERT. We use the best of 3 BERT variants for comparison; i.e., the original BERT, BERT with whole word masking, and BERT without next sentence prediction.

Here, we first compare the performance of BERT and XLNet in a fair setting to decouple the effects of using more data and the improvement from BERT to XLNet. In Table 1, we compare (1) best performance of three different variants of BERT and (2) XLNet trained with the same data and hyperparameters. As we can see, trained on the same data with an almost identical training recipe, XLNet outperforms BERT by a sizable margin on all the considered datasets.

## 3.3 Results After Scaling Up

| RACE | Accuracy | Middle | High | Model | NDCG@20 | ERR@20 |
|---|---|---|---|---|---|---|
| GPT [28] | 59.0 | 62.9 | 57.4 | DRMM [13] | 24.3 | 13.8 |
| BERT [25] | 72.0 | 76.6 | 70.1 | KNRM [8] | 26.9 | 14.9 |
| BERT+DCMN* [38] | 74.1 | 79.5 | 71.8 | Conv [8] | 28.7 | 18.1 |
| RoBERTa [21] | 83.2 | 86.5 | 81.8 | BERT[†] | 30.53 | 18.67 |
| XLNet | **85.4** | **88.6** | **84.0** | XLNet | **31.10** | **20.28** |

Table 2: Comparison with state-of-the-art results on the test set of RACE, a reading comprehension task, and on ClueWeb09-B, a document ranking task. ∗ indicates using ensembles. † indicates our implementations. "Middle" and "High" in RACE are two subsets representing middle and high school difficulty levels. All BERT, RoBERTa, and XLNet results are obtained with a 24-layer architecture with similar model sizes (aka BERT-Large).

After the initial publication of our manuscript, a few other pretrained models were released such as RoBERTa [21] and ALBERT [19]. Since ALBERT involves increasing the model hidden size from 1024 to 2048/4096 and thus substantially increases the amount of computation in terms of FLOPs, we exclude ALBERT from the following results as it is hard to lead to scientific conclusions. To obtain relatively fair comparison with RoBERTa, the experiment in this section is based on full data and reuses the hyper-parameters of RoBERTa, as described in section 3.1.

The results are presented in Tables 2 (reading comprehension & document ranking), 3 (question answering), 4 (text classification) and 5 (natural language understanding), where XLNet generally outperforms BERT and RoBERTa. In addition, we make two more interesting observations:

| SQuAD2.0 | EM | F1 | SQuAD1.1 | EM | F1 |
|---|---|---|---|---|---|
| *Dev set results (single model)* | | | | | |
| BERT [10] | 78.98 | 81.77 | BERT† [10] | 84.1 | 90.9 |
| RoBERTa [21] | 86.5 | 89.4 | RoBERTa [21] | 88.9 | 94.6 |
| XLNet | **87.9** | **90.6** | XLNet | **89.7** | **95.1** |
| *Test set results on leaderboard (single model, as of Dec 14, 2019)* | | | | | |
| BERT* [10] | 80.005 | 83.061 | | | |
| RoBERTa [21] | 86.820 | 89.795 | | | |
| XLNet | **87.926** | **90.689** | | | |

Table 3: Results on SQuAD, a reading comprehension dataset. † marks our runs with the official code. We are not able to obtain the test results on SQuAD1.1 from the organizers after submitting our result for more than one month.

| Model | IMDB | Yelp-2 | Yelp-5 | DBpedia | AG | Amazon-2 | Amazon-5 |
|---|---|---|---|---|---|---|---|
| CNN [15] | - | 2.90 | 32.39 | 0.84 | 6.57 | 3.79 | 36.24 |
| DPCNN [15] | - | 2.64 | 30.58 | 0.88 | 6.87 | 3.32 | 34.81 |
| Mixed VAT [31, 23] | 4.32 | - | - | 0.70 | 4.95 | - | - |
| ULMFiT [14] | 4.6 | 2.16 | 29.98 | 0.80 | 5.01 | - | - |
| BERT [35] | 4.51 | 1.89 | 29.32 | 0.64 | - | 2.63 | 34.17 |
| XLNet | **3.20** | **1.37** | **27.05** | **0.60** | **4.45** | **2.11** | **31.67** |

Table 4: Comparison with state-of-the-art error rates on the test sets of several text classification datasets. All BERT and XLNet results are obtained with a 24-layer architecture with similar model sizes (aka BERT-Large).

| Model | MNLI | QNLI | QQP | RTE | SST-2 | MRPC | CoLA | STS-B | WNLI |
|---|---|---|---|---|---|---|---|---|---|
| *Single-task single models on dev* | | | | | | | | | |
| BERT [2] | 86.6/- | 92.3 | 91.3 | 70.4 | 93.2 | 88.0 | 60.6 | 90.0 | - |
| RoBERTa [21] | 90.2/90.2 | 94.7 | 92.2 | **86.6** | 96.4 | **90.9** | 68.0 | 92.4 | - |
| XLNet | **90.8/90.8** | **94.9** | **92.3** | 85.9 | **97.0** | 90.8 | **69.0** | 92.5 | - |
| *Multi-task ensembles on test (from leaderboard as of Oct 28, 2019)* | | | | | | | | | |
| MT-DNN* [20] | 87.9/87.4 | 96.0 | 89.9 | 86.3 | 96.5 | 92.7 | 68.4 | 91.1 | 89.0 |
| RoBERTa* [21] | 90.8/90.2 | 98.9 | 90.2 | 88.2 | 96.7 | 92.3 | 67.8 | 92.2 | 89.0 |
| XLNet* | 90.9/90.9† | 99.0† | 90.4† | 88.5 | 97.1† | 92.9 | 70.2 | 93.0 | 92.5 |

Table 5: Results on GLUE. ∗ indicates using ensembles, and † denotes single-task results in a multi-task row. All dev results are the median of 10 runs. The upper section shows direct comparison on dev data and the lower section shows comparison with state-of-the-art results on the public leaderboard.

- For explicit reasoning tasks like SQuAD and RACE that involve longer context, the performance gain of XLNet is usually larger. This superiority at dealing with longer context could come from the Transformer-XL backbone in XLNet.

- For classification tasks that already have abundant supervised examples such as MNLI (>390K), Yelp (>560K) and Amazon (>3M), XLNet still lead to substantial gains.

## 3.4 Ablation Study

We perform an ablation study to understand the importance of each design choice based on four datasets with diverse characteristics. Specifically, there are three main aspects we hope to study:

- The effectiveness of the permutation language modeling objective alone, especially compared to the denoising auto-encoding objective used by BERT.

- The importance of using Transformer-XL as the backbone neural architecture.

- The necessity of some implementation details including span-based prediction, the bidirectional input pipeline, and next-sentence prediction.

With these purposes in mind, in Table 6, we compare 6 XLNet-Base variants with different implementation details (rows 3 - 8), the original BERT-Base model (row 1), and an additional Transformer-XL

baseline trained with the denoising auto-encoding (DAE) objective used in BERT but with the bidirectional input pipeline (row 2). For fair comparison, all models are based on a 12-layer architecture with the same model hyper-parameters as BERT-Base and are trained on only Wikipedia and the BooksCorpus. All results reported are the median of 5 runs.

| # | Model | RACE | SQuAD2.0 | | MNLI | SST-2 |
|---|---|---|---|---|---|---|
| | | | F1 | EM | m/mm | |
| 1 | BERT-Base | 64.3 | 76.30 | 73.66 | 84.34/84.65 | 92.78 |
| 2 | DAE + Transformer-XL | 65.03 | 79.56 | 76.80 | 84.88/84.45 | 92.60 |
| 3 | XLNet-Base ($K = 7$) | 66.05 | **81.33** | **78.46** | **85.84/85.43** | 92.66 |
| 4 | XLNet-Base ($K = 6$) | 66.66 | 80.98 | 78.18 | 85.63/85.12 | **93.35** |
| 5 | - memory | 65.55 | 80.15 | 77.27 | 85.32/85.05 | 92.78 |
| 6 | - span-based pred | 65.95 | 80.61 | 77.91 | 85.49/85.02 | 93.12 |
| 7 | - bidirectional data | 66.34 | 80.65 | 77.87 | 85.31/84.99 | 92.66 |
| 8 | + next-sent pred | **66.76** | 79.83 | 76.94 | 85.32/85.09 | 92.89 |

Table 6: The results of BERT on RACE are taken from [38]. We run BERT on the other datasets using the official implementation and the same hyperparameter search space as XLNet. $K$ is a hyperparameter to control the optimization difficulty (see Section 2.3).

Examining rows 1 - 4 of Table 6, we can see both Transformer-XL and the permutation LM clearly contribute the superior performance of XLNet over BERT. Moreover, if we remove the memory caching mechanism (row 5), the performance clearly drops, especially for RACE which involves the longest context among the 4 tasks. In addition, rows 6 - 7 show that both span-based prediction and the bidirectional input pipeline play important roles in XLNet. Finally, we unexpectedly find the the next-sentence prediction objective proposed in the original BERT does not necessarily lead to an improvement in our setting. Hence, we exclude the next-sentence prediction objective from XLNet.

Finally, we also perform a qualitative study of the attention patterns, which is included in Appendix A.6 due to page limit.

## 4 Conclusions

XLNet is a generalized AR pretraining method that uses a permutation language modeling objective to combine the advantages of AR and AE methods. The neural architecture of XLNet is developed to work seamlessly with the AR objective, including integrating Transformer-XL and the careful design of the two-stream attention mechanism. XLNet achieves substantial improvement over previous pretraining objectives on various tasks.

**Acknowledgments**

The authors would like to thank Qizhe Xie and Adams Wei Yu for providing useful feedback on the project, Jamie Callan for providing the ClueWeb dataset, Youlong Cheng, Yanping Huang and Shibo Wang for providing ideas to improve our TPU implementation, Chenyan Xiong and Zhuyun Dai for clarifying the setting of the document ranking task. ZY and RS were supported by the Office of Naval Research grant N000141812861, the National Science Foundation (NSF) grant IIS1763562, the Nvidia fellowship, and the Siebel scholarship. ZD and YY were supported in part by NSF under the grant IIS-1546329 and by the DOE-Office of Science under the grant ASCR #KJ040201.

## Footnotes

[2]To avoid clutter, we omit the implementation details including multi-head attention, residual connection, layer normalization and position-wise feed-forward as used in Transformer(-XL). The details are included in Appendix A.2 for reference.

[3]Hyperparameters for pretraining and finetuning are in Appendix A.4.

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
