[Supplementary Material]

# A  Target-Aware Representation via Two-Stream Self-Attention

## A.1  A Concrete Example of How Standard LM Parameterization Fails

In this section, we provide a concrete example to show how the standard language model parameterization fails under the permutation objective, as discussed in Section 2.3. Specifically, let's consider two different permutations $\mathbf{z}^{(1)}$ and $\mathbf{z}^{(2)}$ satisfying the following relationship

$$\mathbf{z}_{<t}^{(1)} = \mathbf{z}_{<t}^{(2)} = \mathbf{z}_{<t} \quad \text{but} \quad z_t^{(1)} = i \neq j = z_t^{(2)}.$$

Then, substituting the two permutations respectively into the naive parameterization, we have

$$\underbrace{p_\theta(X_i = x \mid \mathbf{x}_{\mathbf{z}_{<t}})}_{z_t^{(1)}=i,\ \mathbf{z}_{<t}^{(1)}=\mathbf{z}_{<t}} = \underbrace{p_\theta(X_j = x \mid \mathbf{x}_{\mathbf{z}_{<t}})}_{z_t^{(1)}=j,\ \mathbf{z}_{<t}^{(2)}=\mathbf{z}_{<t}} = \frac{\exp\left(e(x)^\top h(\mathbf{x}_{\mathbf{z}_{<t}})\right)}{\sum_{x'}\exp\left(e(x')^\top h(\mathbf{x}_{\mathbf{z}_{<t}})\right)}.$$

Effectively, two different target positions $i$ and $j$ share exactly the same model prediction. However, the ground-truth distribution of two positions should certainly be different.

## A.2  Two-Stream Attention

Here, we provide the implementation details of the two-stream attention with a Transformer-XL backbone.

Initial represetation:

$$\forall t = 1, \ldots, T: \quad h_t = e(x_t) \quad \text{and} \quad g_t = w$$

Cached layer-$m$ content represetation (memory) from previous segment: $\tilde{\mathbf{h}}^{(m)}$

For the Transformer-XL layer $m = 1, \cdots, M$, attention with relative positional encoding and position-wise feed-forward are consecutively employed to update the represetntations:

$$\forall t = 1, \ldots, T: \quad \hat{h}_{z_t}^{(m)} = \text{LayerNorm}\left(h_{z_t}^{(m-1)} + \text{RelAttn}\left(h_{z_t}^{(m-1)}, \left[\tilde{\mathbf{h}}^{(m-1)}, \mathbf{h}_{\mathbf{z}_{\leq t}}^{(m-1)}\right]\right)\right)$$

$$h_{z_t}^{(m)} = \text{LayerNorm}\left(\hat{h}_{z_t}^{(m)} + \text{PosFF}\left(\hat{h}_{z_t}^{(m)}\right)\right)$$

$$\hat{g}_{z_t}^{(m)} = \text{LayerNorm}\left(g_{z_t}^{(m-1)} + \text{RelAttn}\left(g_{z_t}^{(m-1)}, \left[\tilde{\mathbf{h}}^{(m-1)}, \mathbf{h}_{\mathbf{z}_{<t}}^{(m-1)}\right]\right)\right)$$

$$g_{z_t}^{(m)} = \text{LayerNorm}\left(\hat{g}_{z_t}^{(m)} + \text{PosFF}\left(\hat{g}_{z_t}^{(m)}\right)\right)$$

Target-aware prediction distribution:

$$p_\theta(X_{z_t} = x \mid \mathbf{x}_{z_{<t}}) = \frac{\exp\left(e(x)^\top g_{z_t}^{(M)}\right)}{\sum_{x'}\exp\left(e(x')^\top g_{z_t}^{(M)}\right)},$$

## A.3  Datasets

### A.3.1  RACE Dataset

The RACE dataset [18] contains near 100K questions taken from the English exams for middle and high school Chinese students in the age range between 12 to 18, with the answers generated by human experts. This is one of the most difficult reading comprehension datasets that involve challenging reasoning questions. Moreover, the average length of the passages in RACE are longer than 300, which is significantly longer than other popular reading comprehension datasets such as SQuAD [29]. As a result, this dataset serves as a challenging benchmark for long text understanding. We use a sequence length of 512 during finetuning.

### A.3.2  SQuAD

SQuAD is a large-scale reading comprehension dataset with two tasks. SQuAD1.1 [30] contains questions that always have a corresponding answer in the given passages, while SQuAD2.0 [29] introduces unanswerable questions. To finetune an XLNet on SQuAD2.0, we jointly apply a logistic regression loss for answerability prediction similar to classification tasks and a standard span extraction loss for question answering [10].

### A.3.3 Text classification Datasets

Following previous work on text classification [39, 23], we evaluate XLNet on the following benchmarks: IMDB, Yelp-2, Yelp-5, DBpedia, AG, Amazon-2, and Amazon-5.

### A.3.4 GLUE Dataset

The GLUE dataset [34] is a collection of 9 natural language understanding tasks. The test set labels are removed from the publicly released version, and all the practitioners must submit their predictions on the evaluation server to obtain test set results. In Table 5, we present results of multiple settings, including single-task and multi-task, as well as single models and ensembles. In the multi-task setting, we jointly train an XLNet on the four largest datasets—MNLI, SST-2, QNLI, and QQP—and finetune the network on the other datasets. Only single-task training is employed for the four large datasets. For QNLI, we employed a pairwise relevance ranking scheme as in [20] for our test set submission. However, for fair comparison with BERT, our result on the QNLI dev set is based on a standard classification paradigm. For WNLI, we use the loss described in [16].

### A.3.5 ClueWeb09-B Dataset

Following the setting in previous work [8], we use the ClueWeb09-B dataset to evaluate the performance on document ranking. The queries were created by the TREC 2009-2012 Web Tracks based on 50M documents and the task is to rerank the top 100 documents retrieved using a standard retrieval method. Since document ranking, or ad-hoc retrieval, mainly concerns the low-level representations instead of high-level semantics, this dataset serves as a testbed for evaluating the quality of word embeddings. We use a pretrained XLNet to extract word embeddings for the documents and queries without finetuning, and employ a kernel pooling network [36] to rank the documents.

### A.4 Hyperparameters

### A.4.1 Pretraining Hyperparameters

| Hparam | Value |
|---|---|
| Number of layers | 24 |
| Hidden size | 1024 |
| Number of attention heads | 16 |
| Attention head size | 64 |
| FFN inner hidden size | 4096 |
| Hidden Dropout | 0.1 |
| GeLU Dropout | 0.0 |
| Attention dropout | 0.1 |
| Partial prediction $K$ | 6 |
| Max sequence length | 512 |
| Batch size | 8192 |
| Learning rate | 4e-4 |
| Number of steps | 500K |
| Warmup steps | 40,000 |
| Learning rate decay | linear |
| Adam epsilon | 1e-6 |
| Weight decay | 0.01 |

Table 7: Hyperparameters for pretraining.

The hyperparameters used for pretraining XLNet are shown in Table 7.

### A.4.2 Hyperparameters for Finetuning

The hyperparameters used for finetuning XLNet on various tasks are shown in Table 8. "Layer-wise decay" means exponentially decaying the learning rates of individual layers in a top-down manner. For example, suppose the 24-th layer uses a learning rate $l$, and the Layer-wise decay rate is $\alpha$, then the learning rate of layer $m$ is $l\alpha^{24-m}$.

| Hparam | RACE | SQuAD | MNLI | Yelp-5 |
|---|---|---|---|---|
| Dropout | | 0.1 | | |
| Attention dropout | | 0.1 | | |
| Max sequence length | 512 | 512 | 128 | 512 |
| Batch size | 32 | 48 | 128 | 128 |
| Learning rate | 2e-5 | 3e-5 | 2e-5 | 1e-5 |
| Number of steps | 12K | 8K | 10K | 10K |
| Learning rate decay | | linear | | |
| Weight decay | | 0.01 | | |
| Adam epsilon | 1e-6 | 1e-6 | 1e-6 | 1e-6 |
| Layer-wise lr decay | 1.0 | 0.75 | 1.0 | 1.0 |

Table 8: Hyperparameters for finetuning.

## A.5 Discussion and Analysis

### A.5.1 Comparison with BERT

To prove a general point beyond one example, we now turn to more formal expressions. Inspired by previous work [37], given a sequence $\mathbf{x} = [x_1, \cdots, x_T]$, we define a set of target-context pairs of interest, $\mathcal{I} = \{(x, \mathcal{U})\}$, where $\mathcal{U}$ is a set of tokens in $\mathbf{x}$ that form a context of $x$. Intuitively, we want the model to learn the dependency of $x$ on $\mathcal{U}$ through a pretraining loss term $\log p(x \mid \mathcal{U})$. For example, given the above sentence, the pairs of interest $\mathcal{I}$ could be instantiated as:

$$\mathcal{I} = \Big\{ \big(x = \text{York}, \mathcal{U} = \{\text{New}\}\big), \ \big(x = \text{York}, \mathcal{U} = \{\text{city}\}\big), \ \big(x = \text{York}, \mathcal{U} = \{\text{New, city}\}\big), \ \cdots \Big\}.$$

Note that $\mathcal{I}$ is merely a virtual notion without unique ground truth, and our analysis will hold regardless of how $\mathcal{I}$ is instantiated.

Given a set of target tokens $\mathcal{T}$ and a set of non-target tokens $\mathcal{N} = \mathbf{x} \backslash \mathcal{T}$, BERT and XLNet both maximize $\log p(\mathcal{T} \mid \mathcal{N})$ but with different formulations:

$$\mathcal{J}_{\text{BERT}} = \sum_{x \in \mathcal{T}} \log p(x \mid \mathcal{N}); \quad \mathcal{J}_{\text{XLNet}} = \sum_{x \in \mathcal{T}} \log p(x \mid \mathcal{N} \cup \mathcal{T}_{<x})$$

where $\mathcal{T}_{<x}$ denote tokens in $\mathcal{T}$ that have a factorization order prior to $x$. Both objectives consist of multiple *loss terms* in the form of $\log p(x \mid \mathcal{V}_x)$. Intuitively, if there exists a target-context pair $(x, \mathcal{U}) \in \mathcal{I}$ such that $\mathcal{U} \subseteq \mathcal{V}_x$, then the loss term $\log p(x \mid \mathcal{V}_x)$ provides a training signal to the dependency between $x$ and $\mathcal{U}$. For convenience, we say a target-context pair $(x, \mathcal{U}) \in \mathcal{I}$ is *covered* by a model (objective) if $\mathcal{U} \subseteq \mathcal{V}_x$.

Given the definition, let's consider two cases:

- If $\mathcal{U} \subseteq \mathcal{N}$, the dependency $(x, \mathcal{U})$ is covered by both BERT and XLNet.
- If $\mathcal{U} \subseteq \mathcal{N} \cup \mathcal{T}_{<x}$ and $\mathcal{U} \cap \mathcal{T}_{<x} \neq \emptyset$, the dependency can only be covered by XLNet but not BERT. As a result, XLNet is able to cover more dependencies than BERT. In other words, the XLNet objective contains more effective training signals, which empirically leads to better performance in Section 3.

### A.5.2 Comparison with Language Modeling

Borrowing examples and notations from Section A.5.1, a standard AR language model like GPT [28] is only able to cover the dependency $(x = \text{York}, \mathcal{U} = \{\text{New}\})$ but not $(x = \text{New}, \mathcal{U} = \{\text{York}\})$. XLNet, on the other hand, is able to cover both in expectation over all factorization orders. Such a limitation of AR language modeling can be critical in real-world applications. For example, consider a span extraction question answering task with the context "Thom Yorke is the singer of Radiohead" and the question "Who is the singer of Radiohead". The representations of "Thom Yorke" are not dependent on "Radiohead" with AR language modeling and thus they will not be chosen as the answer by the standard approach that employs softmax over all token representations. More formally, consider a context-target pair $(x, \mathcal{U})$:

- If $\mathcal{U} \nsubseteq \mathcal{T}_{<x}$, where $\mathcal{T}_{<x}$ denotes the tokens prior to $x$ in the original sequence, AR language modeling is not able to cover the dependency.

- In comparison, XLNet is able to cover all dependencies in expectation.

Approaches like ELMo [27] concatenate forward and backward language models in a shallow manner, which is not sufficient for modeling deep interactions between the two directions.

### A.5.3 Bridging the Gap Between Language Modeling and Pretraining

With a deep root in density estimation[4] [4, 32, 24], language modeling has been a rapidly-developing research area [9, 1, 3]. However, there has been a gap between language modeling and pretraining due to the lack of the capability of bidirectional context modeling, as analyzed in Section A.5.2. It has even been challenged by some machine learning practitioners whether language modeling is a meaningful pursuit if it does not directly improve downstream tasks [5]. XLNet generalizes language modeling and bridges such a gap. As a result, it further "justifies" language modeling research. Moreover, it becomes possible to leverage the rapid progress of language modeling research for pretraining. As an example, we integrate Transformer-XL into XLNet to demonstrate the usefulness of the latest language modeling progress.

### A.6 Qualitative Analysis of Attention Patterns

We compare the attention pattern of BERT and XLNet without finetuning. Firstly, we found 4 typical patterns shared by both, as shown in Fig. 2.

| (a) Content stripes | (b) Local/Self focus | (c) Two segments | (d) Content-based symmetry |

Figure 2: Attention patterns **shared by XLNet and BERT**. Rows and columns represent query and key respectively.

More interestingly, in Fig. 3, we present 3 patterns that only appear in XLNet but not BERT: (a) The self-exclusion pattern attends to all other tokens but itself, probably offering a fast way to gather global information; (b) The relative-stride pattern attends to positions every a few stride apart *relative* to the query position; (c) The one-side masked pattern is very similar to the lower-left part of Fig. 1-(d), with the upper-right triangle masked out. It seems that the model learns not to attend the *relative* right half. Note that all these three unique patterns involve the *relative* positions rather than absolute ones, and hence are likely enabled by the "relative attention" mechanism in XLNet. We conjecture these unique patterns contribute to the performance advantage of XLNet. On the other hand, the proposed permutation LM objective mostly contributes to a better data efficiency, whose effects may not be obvious from qualitative visualization.

| (a) Self exclusion | (b) Relative stride | (c) One-side masked |

Figure 3: Attention patterns that **appear only in XLNet**. Rows and columns represent query and key respectively.

Figure 4: Illustration of the permutation language modeling objective for predicting $x_3$ given the same input sequence $\mathbf{x}$ but with different factorization orders.

## A.7   Visualizing Memory and Permutation

In this section, we provide a detailed visualization of the proposed permutation language modeling objective, including the mechanism of reusing memory (aka the recurrence mechanism), how we use attention masks to permute the factorization order, and the difference of the two attention streams.

As shown in Figure 5 and 6, given the current position $z_t$, the attention mask is decided by the permutation (or factorization order) $\mathbf{z}$ such that only tokens the occur before $z_t$ in the permutation can be attended; i.e., positions $z_i$ with $i < t$. Moreover, comparing Figure 5 and 6, we can see how the query stream and the content stream work differently with a specific permutation through attention masks. The main difference is that the query stream cannot do self-attention and does not have access to the token at the position, while the content stream performs normal self-attention.

Joint View of the Content Stream
(Factorization order: 3 → 2 → 4 → 1)

**Split View**

Position-3 View

Position-2 View

Position-4 View

Position-1 View

Split View of the Content Stream
(Factorization order: 3 → 2 → 4 → 1)

Figure 5: A detailed illustration of the **content stream** of the proposed objective with both the joint view and split views based on a length-4 sequence under the factorization order [3, 2, 4, 1]. Note that if we ignore the query representation, the computation in this figure is simply the standard self-attention, though with a particular attention mask.

Figure 6: A detailed illustration of the **query stream** of the proposed objective with both the joint view and split views based on a length-4 sequence under the factorization order [3, 2, 4, 1]. The dash arrows indicate that the query stream cannot access the token (content) at the same position, but only the location information.

## Footnotes

[4]The problem of language modeling is essentially density estimation for text data.

[5]https://openreview.net/forum?id=HJePno0cYm