[Reviews · NeurIPS 2019]

Reviewer 1



Well motivated, well-executed demonstration of permutation based language modeling for both autoregressive and bidirectional purposes The permutation strategy is new and solves an important problem around the split between autoregressive and bidirectional models This is a complete work, but I would have liked more analysis demonstrating how the new methods change the model dynamics, attention patterns over memory and context, or some empirical attempt at explaining how the new methods create better representations. It is clear and well-organized Many are likely to rely on these ideas or similar ones. This does belong in the thread of literature that succeeds BERT. Update: Thank you for including the attentional analysis. If you have the opportunity to examine which tasks are affected most by transfer and can shed light on which examples are correct by XLNet and not by BERT, that might provide valuable insight into what these pretraining procedures are learning.

Reviewer 2



The paper proposes a clever solution to bridge the gap between training and fine-tuning a language model (i.e. BERT with masked LM objective). The proposed permutation LM allows the model to look at both left and right context, which has been shown to be beneficial for downstream NLP tasks. To achive this, the paper proposed two-stream self-attention, in which one self-attention layer prevent the current word attend to itself (thus makes optimization become trivial, the model just learn to copy). The authors evaluate their model on downstream NLP tasks including some challenging tasks such as RACE M/H, Yelp-5, SQuaD 2.0. In all experiments, XLNet is fairly compared with BERT (Large and Base) and the results show that XLNet performs better than BERT on downstream tasks. The paper is well-written and easy to follow.

Reviewer 3



Originality: The architecture is novel compare to recent lines of language model work, which all used variation of BERT or GPT (SciBERT, MT-DNN, MASS and etc). The example ("New York is a city" one) makes sense, but considering the permutation is random when computing the objective function, I still couldn't get why it works better than sequential order because human speaks/writes in sequential order. Could you add more intuitions in paper? Or have you tried predicting n-gram, compare to permutation? Quality: Very high considering they did extensive of studies on multiple benchmarks, also the ablation study is nicely done as well. However, the comparison is little bit of unfair since BERT only predicts sub-tokens originally. So I hope you could compare to BERT whole word masking (released around end of May) under the same training corpus, if possible. Clarity: The paper is well written and organized. Significance: Achieving first place on multiple benchmarks and more than 3500 stars on their repo explains the significance of this work. I hardly believe there will be lots of follows up on this work because probably only small number of people can afford to train this. But people will use it as the base architecture and fine-tune on top of it, which still benefits the whole NLP community. I am very satisfied with their new ablation studies and will increase my score.

[Author Response · NeurIPS 2019]

We thank all the reviewers for helpful suggestions. We will incorporate the following analysis into our revision.

**(R1) Qualitative analysis via attention patterns.** We compared the attention pattern of BERT and XLNet without finetuning. Firstly, we found 4 typical patterns shared by both, as shown in Fig. 1.

(a) Content stripes     (b) Local/Self focus     (c) Two segments     (d) Content-based symmetry

Figure 1: Attention patterns **shared by XLNet and BERT**. Rows and columns represent query and key respectively.

More interestingly, in Fig. 2, we present 3 patterns that only appear in XLNet but not BERT: (a) The self-exclusion
pattern attends to all other tokens but itself, probably offering a fast way to gather global information; (b) The relative-
stride pattern attends to positions every a few stride apart *relative* to the query position; (c) The one-side masked pattern
is very similar to the lower-left part of Fig. 1-(d), with the upper-right triangle masked out. It seems that the model
learns not to attend the *relative* right half. Note that all these three unique patterns involve the *relative* positions rather
than absolute ones, and hence are likely enabled by the "relative attention" mechanism in XLNet. We conjecture these
unique patterns contribute to the performance advantage of XLNet. On the other hand, the proposed permutation LM
   objective mostly contributes to a better data efficiency, whose effects may not be obvious from qualitative visualization.

(a) Self exclusion     (b) Relative stride     (c) One-side masked

Figure 2: Attention patterns that **appear only in XLNet**. Rows and columns represent query and key respectively.

**(R2 & R3) Fair comparison with BERT-Large (incl. whole word masking).** A fair comparison between XLNet-
Large and the best of three BERT-Large variants, all trained on Wiki + Books only, is presented in Table 1. As we can
   see, XLNet always improves performance, which is consistent with our early ablation on base models.

| Dataset | XLNet-Large-wikibooks | BERT-Large-wikibooks[†] |
|---|---|---|
| SQuAD1.1 (EM/F1) | 88.2/94.0 | 86.7/92.8 |
| SQuAD2.0 (EM/F1) | 85.1/87.8 | 82.8/85.5 |
| RACE | 77.4 | 75.1 |
| MNLI | 88.4 | 87.3 |
| QNLI | 93.9 | 93.0 |
| QQP | 91.8 | 91.4 |
| RTE | 81.2 | 74.0 |
| SST-2 | 94.4 | 94.0 |
| MRPC | 90.0 | 88.7 |
| CoLA | 65.2 | 63.7 |
| STS-B | 91.1 | 90.2 |

Table 1: [†]For BERT, we report the best result among 3 variants including the original BERT, BERT with whole word masking, and BERT without NSP loss.

Figure 3: Performance at different training steps.

**(R2) Performance progress.** In Fig. 3, we plot the finetuning performances of 4 typical tasks at different steps.
Although the performance at 100K is already decent, it keeps improving and does not fully converge at the end of 500K.

**(R2) Other datasets and baselines.** Since the SuperGLUE is released after the NeurIPS deadline, we haven't got a
chance to check it out. We are planning to look into it. The GPT result is copied from the original GPT-1 paper. For
GPT-2 medium, we suspect the performance will not reach SOTA since it wasn't trained to capture bi-directional info.

**(R3) Sequential vs permutation.** Intuitively, the model with permutation can see both $p(\text{new}|\text{york}, \text{is a city})$ and
$p(\text{york}|\text{new}, \text{is a city})$ in expectation, which enables the model to capture bi-directional relations.

[Meta-Review · NeurIPS 2019]

The paper proposes XLNet, a generalized autoregressive pretraining method for language representation learning. The paper shows that XLNet outperforms the state of the art method of BERT on 12 tasks. The paper is of high quality in terms of clarity, technical soundness, significance, and novelty. The authors successfully addressed the issues pointed out by the reviewers. The reviewers are very satisfied with the response.